# LINT: LLM Interaction Network for Clinical Trial Outcome Prediction

Chufan Gao
Department of Computer Science,
University of Illinois
Urbana-Champaign
Urbana, Illinois, USA
chufan2@illinois.edu

Tianfan Fu
Department of Computer Science,
Rensselaer Polytechnic Institute
Troy, New York, USA
futianfan@gmail.com

Jimeng Sun
Department of Computer Science,
Carle Illinois College of Medicine,
University of Illinois
Urbana-Champaign
Urbana, Illinois, USA
jimeng@illinois.edu

## ABSTRACT

Clinical trial outcome prediction aims to predict the success probability of a clinical trial that reaches its desirable endpoint. Most of the effort focuses on developing machine learning models for making accurate predictions with diverse data sources, including clinical trial descriptions, drug molecules, and target disease conditions. Accurate trial outcome prediction helps trial planning and asset portfolio prioritization. Previous works have focused on small-molecule drugs; however, biologics are a quickly growing intervention type that lacks information that is traditionally known for drugs, like molecular properties. Additionally, traditional methods like graph neural networks are much more difficult to apply to biologics data which are a fast-growing type of drug. To address these points, we propose an LLM-based Interaction Network (LINT), a novel method for trial outcome prediction using only free-text descriptions. We validate the effectiveness of LINT with thorough experiments across three trial phases. Specifically, LINT obtains 0.770, 0.740, and 0.748 ROC-AUC scores on phase I, II, and III, respectively, for clinical trials with biologic interventions.

## CCS CONCEPTS

• **Computing methodologies** → **Information extraction**.

## KEYWORDS

Clinical Trial Outcome Prediction, Healthcare, Clinical Trial, Drug Discovery

**ACM Reference Format:**
Chufan Gao, Tianfan Fu, and Jimeng Sun. 2018. LINT: LLM Interaction Network for Clinical Trial Outcome Prediction. In *Proceedings of KDD AI and Data Science for Healthcare Workshop (KDD '24)*. ACM, New York, NY, USA, 13 pages. https://doi.org/XXXXXXX.XXXXXXX

## 1 INTRODUCTION

Accurate estimation of a clinical trial's success probability is essential for stakeholders such as researchers, biopharma investors, and others, informing their scientific and investment decisions. Inaccurate risk evaluation can lead to grave mistakes in drug development choices [52]. Moreover, given the high costs and generally low success rates of trials, it is crucial to prioritize correctly. For example, approval rates for oncology drugs that enter clinical development are estimated to be as low as 3.4-19.4%, 8.7-25.5% for Cardiovascular, 8.2-15% for Central Nervous System, etc [4, 10, 52, 53].

Drug research usually involves two phases: drug discovery and drug development. The goal of drug discovery is to design diverse and novel drug molecular structures with desirable pharmaceutical properties, while the goal of drug development is to evaluate the effectiveness and safety of the drug on human bodies via clinical trials.

A drug needs to pass three phases of clinical trials to be approved and enter the medical market. Specifically, Phase I trials mainly focus on the safety and dosage of the drug molecules to human bodies (20 to 80 participants, several months, 70% of drugs pass this phase), Phase II focuses on efficacy and side-effects (100 to 300 participants, could take several months to 2 years, 33% of drugs pass this phase), and Phase III focuses on efficacy and monitoring of adverse reactions on broader population in treating disease (200 to 3,000 participants, 1 to 4 years, 25-30% of drugs pass this phase) [8].

From a financial perspective, creating a novel drug-based treatment typically requires around 13–15 years and more than 2 billion dollars in research and development [3, 11, 33, 42]. From 2009 to 2018, the FDA approved 355 new drugs and biologics. As a concrete example to see the scale of the cost it takes to bring a drug to market: to treat heart failure, Novartis sponsored phase III of Entresto (a small-molecule drug), which recruited 4822 patients and spanned five years (2014-2019) which eventually yielded results that beneficial, but not statistically significant effects [1] [23, 29, 44, 45, 47]. If machine learning models can predict the approval rate of a clinical trial before it starts, we could circumvent running high-risk trials that are likely to fail, which would save a significant amount of time and resources.

Some of the main challenges for building accurate machine learning models for predicting trial outcomes are **1) limited training data and 2) diverse trial types**. Although clinical trial summaries are publicly available on clinicaltrials.gov, the limited number of labeled trials (success and failure) may be insufficient for training sophisticated machine learning models. Additionally, clinical trial descriptions vary significantly, with biologics often lacking

---

[1] https://clinicaltrials.gov/ct2/show/study/NCT01920711

typical molecular property information compared to conventional small-molecule drugs.

Despite these limitations, there are numerous opportunities for machine learning in this space. For example, a plethora of unstructured text is available that describes the various aspects of the trial and the drugs. Additionally, there also exists literature that describes the pharmaceutical properties of drug molecules (Absorption, Toxicity, etc).

In recent years, there have been numerous breakthroughs in the natural language processing (NLP) community for clinical applications. For example, models can learn semantic knowledge from massive unlabeled data [27], classify unstructured medical text with no human labels [17], process clinical trial tabular data [48, 49], and more [46]. Inspired by recent trends in the NLP space, we propose a novel methodology: LLM-based Interaction Network (LINT). Our model builds on pretrained language models (PLM) to predict trial outcomes by jointly considering the text descriptions of the trial, its associated drugs, and the corresponding medical codes.

We formally define LINT, a deep learning framework for clinical trial outcome prediction that can predict outcomes on *both* small molecule drugs and biologics on Phase I, II, and III clinical trials taken from the largest labeled trial dataset. It leverages text features and International Classification of Diseases (ICD) Codes to accurately predict the approval of interventional trials with *both* small-molecule drugs and biologics. These text features include trial eligibility criteria, trial design specifications, drug property descriptions, and implicit PLM knowledge. In short, LINT learns a function on a weighted combination of pretrained large language model (LLM) embeddings to predict clinical trial outcomes.

- **Experimental results**. The proposed method achieves state-of-the-art performance and beats traditional models. Specifically, LINT obtains ROC-AUC scores of 0.723, 0.702, 0.770 (and F1 scores of 0.643, 0.654, 0.740) on phase I, II, and III, respectively for clinical trials with biologics interventions, which are significantly better than other baselines (Section 3).
- **Open Source Code.** LINT is an open-source, flexible framework that is built on top of pretrained language models (PLM); thus, it is easily adapted when a novel PLM is released. The code will be publicly available at https://github.com/chufangao/LINT
- **Interpretability and Validation**. Although LINT is a deep neural network, we may use Shapley values (a measure of feature importance based on the average contribution of that word to the final prediction probability) to interpret its decisions and visualize which portions of the input text are the most significant [20]. Furthermore, we will show in later sections that LINT's predicted score generally corresponds to the actual approval rate (see Section 3.2).

## 1.1 Related Work on Trial Outcome Prediction

Existing works often focus on predicting individual patient outcomes in a trial instead of a general prediction about the overall trial success. They usually leverage expert, hand-crafted features. Previous work has taken advantage of many aspects of the trial-related features [18, 21, 28, 39, 43, 54, 55].

Deep learning techniques have also seen widespread usage in learning representations from clinical trial data to support downstream tasks such as drug repurposing [2, 14–16, 57], patient retrieval [12, 19, 41] and enrollment [1, 56]. Still, none have fully integrated the textual features along with the tabular trial information (such as allocation, primary purpose, and more) in a natural way.

Additionally, unlike previous work, the dataset used to train and test our proposed model is much larger than before, including both small molecule trials and biologic trials, extending the previous dataset used by Fu et al. [12, 13]. Also, these works generally optimize the representation learning for a single component in a trial, whereas LINT models a diverse set of trial text and tabular data jointly. Furthermore, previous methods will not work for biologics, which lack information that small-molecule drugs generally possess. [30] enhance the accuracy of [13] by quantifying uncertainty of prediction.

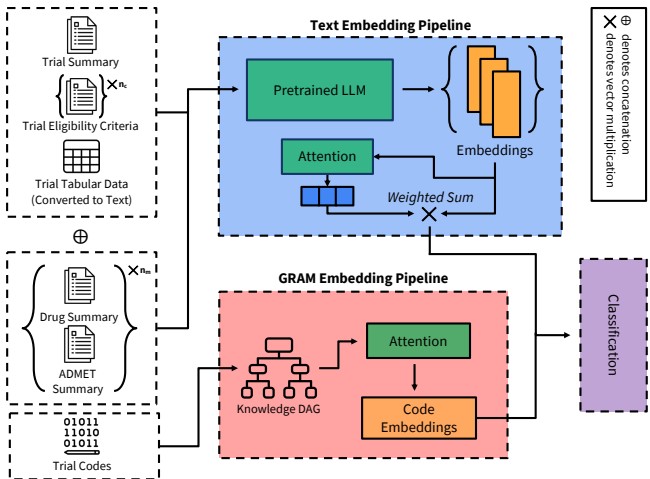

**Figure 1: Structure of proposed LINT model. The input is a series of free texts corresponding to the clinical trials as well as their associated drug / biologics interventions. All the text is encoded by the pretrained LLM, and the resulting embeddings are further processed by a transformer encoder. $n_c$ is the number of eligibility criteria, $n_m$ is the number of molecules, and $n_d$ is the number of ICD codes for the respective input trial. We treat the tabular data as additional sentences for the LLM model. See Section 2 for details. The ICD codes associated with the clinical trials are encoded by GRAM (Graph-based Attention Model for Healthcare Representation Learning), a hierarchical, attention-based method. Finally, the final feedforward neural network performs the classification task.**

## 2 METHODOLOGY

### 2.1 Formulation and Data Featurization

In this section, we formulate the clinical trial outcome prediction problem into a binary classification problem. Additionally, we review all the data features, including trial eligibility criteria, drug descriptions, and disease codes.

Let the set of drug or biological interventions be denoted as:

$$M = \{m_1, m_2, \ldots, m_{N_M}\},$$

where $N_M$ is the total number of distinct interventions.

Each drug $m_i$ = {*description, pharmacodynamics, toxicity, metabolism, absorption*} also has plain text information regarding the ADMET properties and more (retrieved via DrugBank [50, 56])[2]:

- *description:* Brief summary about the drug–primary use cases, symptoms it treats;
- *pharmacodynamics:* Description of how the drug works at a clinical or physiological level;
- *toxicity:* Lethal dose (LD50) values from test animals, description of side effects and toxic effects seen in humans;
- *metabolism:* Mechanism by which or organ location where the drug is neutralized;
- *absorption:* Description of how much of the drug or how readily the drug is taken up by the body;

Let the set of disease codes be denoted as

$$D = \{d_1, d_2, \ldots, d_{N_D}\},$$

where $N_D$ is the total number of distinct diseases, and each $d_i$ is an ICD code.

Let each trial be represented as

$$T = \{c_i, \ldots, c_{n_c}, sum, d_i, \ldots d_{n_d}, m_i, \ldots, m_{n_m}, \mathbf{x}\},$$

where $c_i$ is the *ith* inclusion/exclusion criterion sentence, *sum* is the text summary of the trial, $d_i$ is the *ith* associated ICD code, $m_i$ is the *ith* associated drug/biologic intervention, and

$\mathbf{x}$ = {*ec_gender, ec_min_age, ec_max_age, allocation, intervention_model, primary_purpose, masking, sponsors, continent*}

represents a feature vector of tabular features in the trial, described in the following section.

**Tabular Trial Data:** Specifically, input is a list of text. First, each sentence $c_i$ is separated out in the trial criteria $\{c_i, \ldots, c_{N_c}\}$ by splitting into new lines in the raw trial criteria text; these sentences are added to the list. Next, the brief summary information is appended to the list. Finally, additional tabular table information is processed and combined with the previous list (details in Appendix A.5).

Note that each of the categorical variables also has an additional value for missing values. Finally, each of these categorical features is converted into text by the simple function template *linearization*($\mathbf{x}$) = "[feature_name] [feature_value]; ...; [feature_name] [feature_value]".

**Data Processing for ICD Codes:** To obtain a list of relevant ICD Codes, we use the API provided by https://clinicaltables.nlm.nih.gov/ by inputting the conditions associated with each trial.

### 2.2 Neural Architecture

LINT learns a classifier $\hat{y} = f_\theta(T)$, where $T$ is the trial with the associated drug intervention information as above. More specifically, LINT has 2 main modules, the transformer module, and the GRAM module.

$$f_\theta(T) = f_{\theta'}(CONCAT(h_{text}, h_{GRAM})),$$

where $f_{\theta'}$ is a classifier trained on the output of LINT's inner components, a transformer model that obtains embeddings

$$h_{text} = f_{\theta^{text}}(c_i, \ldots, c_{n_c}, sum, m_i, \ldots, m_{n_m}, x)$$

and a GRAM model that obtains embeddings respectively

$$h_{GRAM} = \frac{1}{n_d} \sum_{i=1}^{n_d} GRAM(d_i)$$

for a specific trial. Note that indexes for the arbitrary number of codes, eligibility criteria, and interventions are omitted.

**LINT Encoder: Text Data:** This module handles all the text information in our input data and outputs embedding $h_{text}$. We use a pretrained BERT model to extract the text features. BERT (Bidirectional Encoder Representations from Transformers) is a pretraining technique that captures language semantics and exhibits state-of-the-art performance in various NLP tasks [9]. Specifically, we use BioBERT [27] as implemented in the HuggingFace library [51]. BioBERT is pretrained on biomedical corpora–namely–PubMed abstracts and PubMed Central (PMC) full-text articles. This offers higher performance on domain-specific tasks such as biomedical relation extraction.

Due to the small number of trials, we use the transformer encoder to obtain a weighted mean over the original BioBERT embeddings. This allows LINT to take advantage of the powerful PLM while adding an additional layer of attention.

Specifically, $f_{\theta^{text}}$ is consists of a PLM $LLM()$ and a transformer encoder $ENCODER()$. $LLM()$ simply takes in a varying-length text input and outputs a 768-dimensional embedding. $ENCODER()$ is a transformer encoder[3] combined with a simple Linear layer that takes in a varying-length list of 768-dimensional inputs and outputs a scalar attention weight for all inputs, i.e.,

$$E = LLM(c_i, \ldots, c_{n_c}, sum, m_i, \ldots, m_{n_m}, x)$$

$$W = ENCODER(E)$$

$$f_{\theta^{text}} = E \cdot W,$$

where $E \in R^{B \times n}, W \in R^n$. $B$ is 768 in this case, and $n = n_c + 1 + n_m + len(x)$, i.e., the total length of the list of text inputs where $n_c$ is the number of eligibility criteria, 1 represents the summary embedding, $n_m$ is the number of molecules, and $len(x)$ the length of the tabular data of the specific trial.

**GRAM Encoder: Disease Code Representation:** This module handles all the ICD code information in our input data outputs embedding $h_{GRAM}$.

Disease codes are typically hierarchically organized in a directed tree. For example, ICD-10 [36, 53] codes consist of three to seven characters. The beginning 3 characters represent the category to which the code belongs to: e.g., the range of A00-B99 consists of "Certain Infectious and Parasitic Diseases". The latter characters

---

[2]https://dev.drugbank.com/guides/fields/drugs

[3]https://pytorch.org/docs/stable/generated/torch.nn.TransformerEncoder.html

represent the specific condition: e.g., A15.0 represents "Tuberculosis of lung".

We leverage graph attention-based model (GRAM) [7] to represent disease code. Specifically, suppose $d$ is the disease code of interest, the ancestors of $d$ are $\mathcal{D}$. Then, GRAM yields an embedding for a given ICD code by the formulation: $GRAM(d) = \sum_{d_j \in \mathcal{D} \cup \{d\}} \alpha_j d_j$, where $\alpha_j$ are the learned attention weights for each level of ICD code.

Cross entropy loss is leveraged to guide the training

$L = \sum_{i=1}^{N} -y \log \hat{y}$, where $y \in \{0, 1\}$ and $\hat{y} \in (0, 1)$ denote groundtruth and prediction, respectively.

**LINT Classification:** Finally, the outputs of the 2 previous subsections are concatenated and input to a multilayer perceptron (MLP) for binary classification. Because our data is imbalanced, we use a weighted binary-cross-entropy loss, where the weight is calculated based on the label distribution in the training data. $\ell(x, y) = -\sum_{c=1}^{2} w_c \log \hat{y}_c y_c$ Where $y_i$ is the predicted probability of class $i$ and $w_i$ is the class weight of $i$.

## 3 EXPERIMENTS

In this section, we present the experimental results. We start by describing the experimental setup, including dataset and splitting (Section 3.1). Finally, we present the trial outcome prediction results and analysis (Section 3.2).

### 3.1 Data

All the historical clinical trial records can be downloaded from https://clinicaltrials.gov/. The trial success information is based on the benchmark from Fu et al. [13][4].

**Table 1: A breakdown of the trials after preprocessing of the 426,368 found on clinicaltrials.gov. No Label indicates the lack of success or failure label found. Non-Biological or Small-Molecule indicates that the drug does not have either of these interventions (i.e., those clinical trials may involve medical devices and behavioral interventions). We follow the labels and data splits provided by Fu et al. [13]. A further breakdown of the labeled data is shown in Table 2.**

| Flag | Number of trials |
|---|---|
| No Label | 149,091 |
| Non-Biological or Small-Molecule | 155,353 |
| Non Interventional | 96,538 |
| Labeled Trials | 25,386 |

**Trial data preprocessing:** Data preprocessing pared down the original total of 426,368 clinical trials to 23,519 valid trials for consideration. We focused on interventional trials, excluding observational ones, and further narrowed the scope to biological or drug interventions. Trials with efficacy concerns or lacking outcome labels were also omitted. The final breakdown comprised of (4437 Phase I, 11214 Phase II, and 7868 Phase III trials). See Table 1 for a detailed account of the number of trials post-preprocessing.

[4]https://github.com/futianfan/clinical-trial-outcome-prediction

**Table 2: Table showcasing data partitions according to modality and phase. The final two columns provide the total quantity of training data for training and testing, and the percentage of successful trials—in parentheses—of all trials. Bio. refers to biologics, Drugs to small-molecule drug candidates, and Both to a combined set of biologic and small-molecule trials.**

| Mode | Phase | # Train (Pos. %) | # Test (Pos. %) |
|---|---|---|---|
| Bio. | 1 | 505 (71.29%) | 441 (71.88%) |
| | 2 | 973 (59.61%) | 571 (55.52%) |
| | 3 | 692 (78.03%) | 366 (74.32%) |
| Drugs | 1 | 2032 (59.45%) | 1681 (59.79%) |
| | 2 | 6401 (48.68%) | 3873 (54.40%) |
| | 3 | 4745 (65.99%) | 2388 (67.63%) |
| Both | 1 | 2418 (62.57%) | 2019 (62.95%) |
| | 2 | 6999 (50.49%) | 4215 (55.26%) |
| | 3 | 5249 (67.96%) | 2619 (69.19%) |

**Table 3: Phase 3 baseline comparisons on TOP test set. Drugs\* denotes small-molecule drugs. Combined indicates the combined set of biologics and small molecule drugs.**

| Mode | Model | PR-AUC | ROC-AUC | F1 | Acc |
|---|---|---|---|---|---|
| Bio. | LR | 0.846 ± 0.029 | 0.697 ± 0.043 | 0.859 ± 0.015 | 0.777 ± 0.022 |
| | SVM | 0.867 ± 0.019 | 0.692 ± 0.035 | 0.872 ± 0.014 | 0.785 ± 0.022 |
| | DT | 0.797 ± 0.019 | 0.622 ± 0.028 | 0.799 ± 0.016 | 0.704 ± 0.022 |
| | AB | 0.857 ± 0.022 | 0.705 ± 0.036 | 0.855 ± 0.010 | 0.767 ± 0.014 |
| | RF | 0.863 ± 0.016 | 0.731 ± 0.022 | 0.844 ± 0.012 | 0.755 ± 0.017 |
| | LINT | **0.882 ± 0.016** | **0.770 ± 0.028** | **0.879 ± 0.010** | **0.817 ± 0.016** |
| Drugs* | LR | 0.831 ± 0.011 | 0.703 ± 0.010 | 0.781 ± 0.010 | 0.691 ± 0.011 |
| | SVM | 0.811 ± 0.011 | 0.681 ± 0.012 | 0.797 ± 0.009 | 0.692 ± 0.012 |
| | DT | 0.717 ± 0.010 | 0.573 ± 0.010 | 0.717 ± 0.008 | 0.621 ± 0.009 |
| | AB | 0.814 ± 0.012 | 0.692 ± 0.013 | 0.794 ± 0.007 | 0.700 ± 0.009 |
| | RF | 0.760 ± 0.013 | 0.635 ± 0.011 | 0.756 ± 0.009 | 0.658 ± 0.010 |
| | HINT | 0.733 ± 0.009 | 0.691 ± 0.014 | 0.792 ± 0.007 | 0.695 ± 0.008 |
| | LINT | **0.854 ± 0.010** | **0.740 ± 0.011** | **0.820 ± 0.008** | **0.726 ± 0.011** |
| Both | LR | 0.856 ± 0.009 | 0.732 ± 0.010 | 0.806 ± 0.007 | 0.717 ± 0.008 |
| | SVM | 0.833 ± 0.007 | 0.699 ± 0.012 | 0.808 ± 0.005 | 0.705 ± 0.007 |
| | DT | 0.724 ± 0.010 | 0.563 ± 0.013 | 0.720 ± 0.009 | 0.619 ± 0.011 |
| | AB | 0.832 ± 0.009 | 0.704 ± 0.010 | 0.800 ± 0.007 | 0.705 ± 0.009 |
| | RF | 0.781 ± 0.011 | 0.647 ± 0.013 | 0.769 ± 0.006 | 0.670 ± 0.007 |
| | LINT | **0.860 ± 0.009** | **0.748 ± 0.009** | **0.826 ± 0.005** | **0.737 ± 0.007** |

**Table 4: Results of LINT on the different data splits on the TOP test set. Drugs\* denotes small-molecule drugs.**

| Mode | Phase | PR-AUC | ROC-AUC | F1 | Acc. |
|---|---|---|---|---|---|
| Bio. | 1 | 0.860 ± 0.026 | 0.723 ± 0.029 | 0.778 ± 0.015 | 0.694 ± 0.018 |
| | 2 | 0.758 ± 0.022 | 0.702 ± 0.011 | 0.687 ± 0.016 | 0.651 ± 0.011 |
| | 3 | 0.882 ± 0.016 | 0.770 ± 0.028 | 0.879 ± 0.010 | 0.817 ± 0.016 |
| Drugs* | 1 | 0.728 ± 0.014 | 0.643 ± 0.014 | 0.698 ± 0.008 | 0.615 ± 0.009 |
| | 2 | 0.696 ± 0.010 | 0.654 ± 0.007 | 0.678 ± 0.008 | 0.606 ± 0.008 |
| | 3 | 0.854 ± 0.010 | 0.740 ± 0.011 | 0.820 ± 0.008 | 0.726 ± 0.011 |
| Both | 1 | 0.770 ± 0.015 | 0.667 ± 0.013 | 0.716 ± 0.010 | 0.637 ± 0.010 |
| | 2 | 0.699 ± 0.010 | 0.650 ± 0.006 | 0.706 ± 0.006 | 0.585 ± 0.007 |
| | 3 | 0.860 ± 0.009 | 0.748 ± 0.009 | 0.826 ± 0.005 | 0.737 ± 0.007 |

**Prediction by trial phase:** Predictions are reported by trial phase, reflecting the distinct objectives of each phase. Phase I primarily determines the safe maximum dosage, focusing on potential adverse

effects. Phase II examines the efficacy, using metrics tailored to the intervention's aim - such as tumor size, survival rate, or quality of life. This phase typically experiences the most noise, given the complex task of defining success. Phase III compares the new drug with existing treatments, often employing double-blind methodologies involving multiple treatment arms.

**Train/Test Split:** To split the data into train, test, and validation sets, we manually chose the year 2015 as the cutoff date (specifically, January 1, 2015), following the methodology of [5, 13, 14]. This split strategy is to circumvent the information leakage because later trials are typically relying on the knowledge from the earlier trials. All trials whose completion date is confirmed to be before the cutoff are considered training and validation data; otherwise, they are considered test data.

Finally, we also split by Phase and Intervention Type. We consider predicting trial outcomes from phases 1, 2, and 3. There are 3 interventions that we consider: only small molecule drugs, only biologics, or both. Table 2 shows the full information regarding splits based on drug type and phase.

Information regarding baselines–Logistic Regression (LR), Support Vector Machine (SVM), Decision Tree (DT), AdaBoost (AB), Random Forest (RF), and HINT–are shown in Appendix A.4.3.

### 3.2 Results & Analysis

In this section, we present and analyze the experimental results. Table 4 reveals that `LINT` excels at Phase 3 prediction, garnering Test AUC scores of 0.770, 0.740, and 0.745 for biologic, drugs, and combined predictions, respectively. While the model delivers a solid performance in Phase 1 prediction, its overall performance dips in Phase 2. This downturn is expected, given that Phase 2 has the highest trial volume and is generally the most challenging task.

Table 3 comparison reveals `LINT` surpasses all baseline models in all metrics. The simple Logistic Regression with our BERT embedding input, the second-best model, achieves Phase 3 Test F1 scores of 0.865, 0.800, and 0.806, which still lags considerably behind `LINT`. Note that HINT is solely applicable in Drugs mode, as it doesn't cater to Biologics.

**Model Calibration:**

From Figure 2, we can visualize the probability of actual success vs. model-predicted success for phase 3, and from Figure 6 we see the number of predictions by probability. Most of the predictions have higher scores (which makes sense as there are more positive true labels). We generally see a positive correlation between the predicted vs. actual success probability, which confirms that our model is well-calibrated, even without explicit tuning for calibration. One notable exception can be seen with the predictions greater than 0.9. This is because there are very few predictions for biologics that have normalized probabilities larger than 0.9, as shown in Figure 6 (exactly one trial). This could be because of the noise level in the combined labels of all drugs and biologics; regardless, further work should aim to factor in calibration as a metric for model performance to improve human trust. Additionally, we observe that model predictions below 0.4, 0.2, and 0.4 yields entirely true negative predictions.

**Performance Breakdown by Disease Type:** Table 6 shows the top 5 most common ICD categories for each of the three modes (by

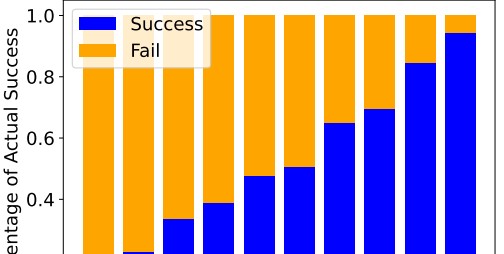

**Figure 2: Phase 3 Predictions of success vs actual success. This can be interpreted as `LINT`'s predicted probability of success in the X-axis (e.g., X=0.1 contains all success probability predictions in the range [0.1,0.2)), versus the actual probability of successful trials among the predicted successful trial on the Y-axis (given the predictions ). Combined refers to the combined.**

occurrence in the clinical trial data). Note that the most common categories are naturally different for each modality. However, we can see that `LINT` generally performs well and achieves high accuracy and ROC-AUC over most categories. It is interesting to see that for Biologics interventions for Neoplasms (cancer-related trials), the F1 score is much lower than the accuracy. This indicates that our model tends to predict true negative samples better than true positives in the Neoplasm case. However, in most other categories, both and the accuracy is high.

## 4 CONCLUSION

Clinical trial outcome prediction is vital for predicting the safety of new drugs and biologics. In this paper, we focus on developing a machine learning model to predict the outcome of clinical trial that can account for biologics, a quickly growing intervention type. Specifically, we propose an open-source, flexible framework that is built on top of pretrained language models–`LINT`–a method that supports the accurate prediction of success in clinical trials.

Thorough empirical studies are carried out to validate the effectiveness of the proposed method, which achieves state-of-the-art ROC-AUC scores on predicting approval of phase III trials, beating many traditional and recent baselines. We validate the effectiveness of `LINT` with thorough experiments across three trial phases. Specifically, `LINT` obtains 0.770, 0.740, and 0.748 ROC-AUC scores on phase I, II, and III, respectively for clinical trials with biologic interventions. We also show that `LINT` is generally well calibrated and demonstrated `LINT`'s performance on the top-5 most popular categories of ICD codes. Additionally, using Shapley values, we visualize the portions of the input text that are the most significant for the prediction of success/failure.

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

# A APPENDIX

## A.1 Future Work

**Future Work:** Future research should address the substantial lack of clear outcome labels in clinical trial datasets, an issue currently unaddressed, with potentially over 100,000+ unlabeled trials, as indicated in Table 1. Unsupervised learning strategies, such as masked language modeling, weak supervision, or semi-supervision, could be instrumental in resolving this. It's also crucial to improve label quality, a challenging task due to the complex language in result descriptions. Endeavors to identify human-interpretable automatic labels could help expand the dataset. Interpretability is also key for decision-makers. Understanding why a trial is predicted to fail or succeed, rather than simply knowing the outcome, is valuable. While LINT can be paired with Shapley values to highlight text sections affecting prediction confidence, it doesn't directly account for human interpretability. Future research should focus on creating interpretable models, which could optimize clinical trial design and enhance success rates.

## A.2 Notations

Table 5 shows relevant notation used in the paper.

## A.3 Additional Tables

In this section, we provide additional Tables that are not featured in the main text due to length limitations. For example, Table 6 shows LINT performance on popular ICD categories.

## A.4 Literature Review Extended

In this section, we briefly review the related literature. In recent years, there have been several attempts to use machine learning to predict clinical trial outcomes.

*A.4.1 Traditional Machine Learning Methods.* Lo et al. [28] applied traditional machine-learning techniques (penalized logistic regression (PLR), random forests (RF), neural networks (NN), gradient boosting trees (GBT), support vector machines (SVM) [37], and decision trees C.50 [26]) to predict drug approvals using drug and clinical trial data. Payvert et al. [21] introduced PrOCTOR, a random forest-based model to predict drug toxicity using 10 molecular properties, 34 target-based properties, and 4 drug-likeness rule features. Hong et al. [24] designed an ensemble classifier based on weighted least squares support vector regression (LS-SVR) to predict the success/failure of clinical trials.

Wu et al. [54] developed a two-stage SVM classification method to identify genes and genetic lesion statuses in clinical trials. Raj et al. [39] used Gradient-Boosted Decision Trees (GBDT [55]) to predict patients who responded to treatment on various depressive symptoms utilizing pretreatment symptom scores and electroencephalographic features. Siah et al. [43] created an open challenge and compared over 3000 various machine learning models for clinical trial outcome prediction. They found that the best-performing model was an ensemble consisting of two XGBoost models [6] and one Bayesian logistic regression (BLR) model [22].

Our proposed model LINT differs from the previously mentioned methods in several ways. Most do not consider drug molecule features and trial protocol information jointly, rather generally

**Table 5: Mathematical notations and their explanations.**

| Notations | Explanations |
|---|---|
| $y \in \{0, 1\}$ | Groundtruth, binary label |
| $c_1, \cdots,$ | inclusion/exclusion criterion sentence |
| $\mathbf{x}$ | A feature vector of tabular data related to the trial |
| $sum$ | Text summary of the trial |
| $D = \{d_1, d_2, \ldots, d_{N_D}\}$ | Set of ICD disease codes |
| $M = \{m_1, m_2, \ldots, m_{N_M}\}$ | Set of drug interventions |
| $T = \{c_1, \ldots, c_{n_c}, sum, d_1, \ldots d_{n_d}, m_1, \ldots, m_{n_m}, \mathbf{x})\}$ | A trial is a set of these variables |
| $h_{text} = f_{\theta^{text}}(c_i, \ldots, c_{n_c}, sum, m_i, \ldots, m_{n_m}, \mathbf{x})$ | Embedding obtained from transformer encoder |
| $E = LLM(c_i, \ldots, c_{n_c}, sum, m_i, \ldots, m_{n_m}, \mathbf{x}))$ | BERT encoder, maps text to embedding space $E \in R^B$ |
| $W = ENCODER(E)$ | Transformer encoder, maps embeddings to attention $R^B \rightarrow R$ |
| $f_{\theta^{text}} = E \cdot W$ | Weighted mean of embeddings |
| $h_{GRAM} = \frac{1}{n_d} \sum_{i=1}^{n_d} GRAM(d_i)$ | Embedding obtained from GRAM model |
| $\hat{y} = f_\theta(T) = f_{\theta'}(CONCAT(h_{text}, h_{GRAM}))$ | Final LINT classifier |

**Table 6: A table of popular ICD categories (taken from the trial's ICD code) for small molecule drugs, biologics, and both combined. N represents the number of samples, and (Hist. App. %) represents the historical approval rate–the positive labels indicating trial successes. All metrics are from LINT. (Note that the category "Factors influencing health status and contact with health services" includes codes that represent HIV infection status, family and personal history of cancer, family and personal history of diabetes, and cystic fibrosis carrier)**

| Mode | Category | N (Hist. App. %) | Test PR-AUC | Test ROC-AUC | Test F1 | Test Acc. |
|---|---|---|---|---|---|---|
| Biologics | Neoplasms | 80 (37.50) | 0.603 ± 0.090 | 0.722 ± 0.048 | 0.435 ± 0.075 | 0.713 ± 0.042 |
| | Certain infections and parasitic diseases | 64 (84.38) | 0.915 ± 0.049 | 0.674 ± 0.123 | 0.913 ± 0.032 | 0.848 ± 0.050 |
| | Factors influencing health status and contact with health services | 47 (72.34) | 0.897 ± 0.061 | 0.870 ± 0.064 | 0.899 ± 0.039 | 0.855 ± 0.054 |
| | Diseases of the respiratory system | 42 (83.33) | 0.943 ± 0.025 | 0.761 ± 0.084 | 0.926 ± 0.029 | 0.877 ± 0.045 |
| | Diseases of the musculoskeletal system and connective tissue | 41 (78.05) | 0.846 ± 0.077 | 0.603 ± 0.114 | 0.880 ± 0.046 | 0.801 ± 0.068 |
| Small-Molecule Drugs | Neoplasms | 568 (53.52) | 0.719 ± 0.029 | 0.709 ± 0.027 | 0.703 ± 0.023 | 0.670 ± 0.022 |
| | Factors influencing health status and contact with health services | 373 (70.51) | 0.900 ± 0.019 | 0.798 ± 0.024 | 0.847 ± 0.014 | 0.765 ± 0.017 |
| | Endocrine, nutritional and metabolic diseases | 345 (79.42) | 0.934 ± 0.022 | 0.787 ± 0.044 | 0.893 ± 0.016 | 0.814 ± 0.025 |
| | Diseases of the nervous system | 322 (62.11) | 0.816 ± 0.022 | 0.723 ± 0.027 | 0.767 ± 0.016 | 0.648 ± 0.020 |
| | Certain infections and parasitic diseases | 313 (73.80) | 0.877 ± 0.032 | 0.751 ± 0.033 | 0.873 ± 0.019 | 0.787 ± 0.029 |
| Combined | Neoplasms | 585 (53.68) | 0.752 ± 0.028 | 0.724 ± 0.023 | 0.672 ± 0.024 | 0.660 ± 0.018 |
| | Factors influencing health status and contact with health services | 405 (71.60) | 0.902 ± 0.014 | 0.793 ± 0.019 | 0.847 ± 0.010 | 0.768 ± 0.013 |
| | Certain infections and parasitic diseases | 360 (75.83) | 0.871 ± 0.021 | 0.730 ± 0.029 | 0.878 ± 0.018 | 0.792 ± 0.028 |
| | Endocrine, nutritional and metabolic diseases | 351 (79.77) | 0.930 ± 0.017 | 0.779 ± 0.031 | 0.893 ± 0.012 | 0.816 ± 0.019 |
| | Diseases of the nervous system | 334 (62.87) | 0.808 ± 0.027 | 0.709 ± 0.026 | 0.770 ± 0.019 | 0.659 ± 0.021 |

relying on sets of hand-annotated features. Furthermore, most trials suffer from small training data sources, whereas LINT takes into account a large, multi-modal dataset of text and tabular data.

*A.4.2 Deep Representation Learning Related to Clinical Trials.* Recently, deep learning has been rising in popularity in the machine learning for healthcare space; specifically, it has been used to learn representation from clinical trial data to support downstream tasks such as drug repurposing [14, 25], patient retrieval [19, 58] and enrollment [1].

Doctor2Vec [1], a recently proposed hierarchical clinical trial embedding where the unstructured trial descriptions were embedded using Bidirectional Encoder Representations from Transformers (BERT) [9].

DeepEnroll [58] leverages a hierarchical embedding model to represent patient longitudinal electronic health record (EHR) and aligns it with eligibility criteria (EC) via a numerical information embedding and entailment module to reason over numerical information in both EC and EHR.

Gao et al. [19] proposed a patient-trial matching model to find qualified patients for clinical trials given structured EHR and unstructured EC text with both inclusion and exclusion criteria. The core of this model consists of a convolutional highway network and a hierarchical memory network that generates a contextualized word embedding for each word of the trial protocol. Multiple one-dimensional convolutional layers with varying kernel sizes capture semantics at different granularity.

Qi et al. [38] designed a Residual Semi-Recurrent Neural Network and took phase 2 results as features to predict the phase 3 outcome. This network consists of an RNN with a residual connection from the first input and performs significantly better than RNNs. The trough concentration (Ctrough) and Phase 2 subject–level baseline characteristics were used to build an individual treatment effect (ITE) model for Phase 3 trial patients.

Fu et al. [13] designed a Hierarchical Interaction Network (HINT) to capture the interaction between multi-modal features (drug molecules, disease codes, eligibility). It uses an interaction graph module on embeddings produced via domain knowledge to capture various relations between EC, molecule structure, trial protocol, and more to predict trial outcomes. However, this work does not support biologics-related interventions because the lack of protein structures and molecule properties such as absorption, distribution, metabolism, excretion, and toxicity (ADMET) are not known.

*A.4.3 Baseline Methods.* We employ the following baseline methods for clinical trial outcome prediction and compare them with our proposed LINT method. Each baseline method has the average text embedding of the input texts as well as the GRAM embeddings of the ICD codes for diseases addressed in the clinical trials.

- **Logistic Regression** is a common model that models the log-odds for a class through a linear combination of the input features, similar to a simple one-layer neural network with a logistic activation function [28].
- **Supporting Vector Machine (SVM)** is another common linear model that attempts to fit a maximum-margin hyperplane between the input features (often using a nonlinear kernel function) in order to separate classes [54].
- **Decision Tree** is a hierarchical, rule-based model that's generally trained using algorithm that attempts to iteratively split on a feature using an information-theoretic measure like label entropy at each branch of the tree [28].
- **AdaBoost** is a meta-estimator that iteratively fits a decision tree and then fits additional copies of the classifier on the re-weighted dataset (where weights of incorrectly classified instances are increased to emphasize them) [13].
- **Random Forest** is an ensemble of decision trees trained on different sub-samples of the input data (usually via sampling with bootstrapping) [28].
- **Hierarchical Interaction Network (HINT)** is the previous state-of-the-art model we compare against. It is a complex model, consisting of a graph attention network, highway networks, and more to combine drug structure, eligibility criteria, and ICD codes in order to make a binary trial success prediction [13].

## A.5 Trial Details
We describe the processing for each of the tabular features in **x**.

- *ec_gender:* The eligible patient genders (male/female / or either) that the trial considers from eligibility criteria.
- *ec_min_age, ec_max_age:* The minimum and maximum ages of patients selected via the eligibility criteria. Notes that we convert the valid age range of the eligibility criteria to 4 bins following the Research Inclusion Statistics Report from the NIH [35]. Ages below 6 are considered children, ages 6-18 are considered adolescents, and ages 18-65 are considered adults. Finally, ages higher than 65 are considered older adults. I.e., 4 bins of (<6, 6-18, 18-65 and >65).
- *allocation:* the treatment allocation, which can be randomized or nonrandomized.
- *intervention_model:* The general design of the strategy for assigning therapies and can be Crossover, Factorial, Parallel, Sequential, or Single Group Assignment.
- *primary_purpose:* Describes the trial purpose, including Basic Science, Diagnostic, Educational/Counseling/Training, Health Services Research, Prevention, Screening, Supportive Care, Treatment, or Other.
- *masking:* The type of method (single, double, triple, or quadruple masking) used to keep the study group assignment hidden after allocation between parties (Participants, Care Providers, Investigators, and Outcomes Assessors).
- *sponsors:* The organization that oversees the trial. Since there are hundreds of possible sponsors, we simply denote separate sponsors into Large or Small, where large sponsors are the top 10 most common sponsors over all trials [5].
- *continents:* The continents in which the study was performed were converted from the raw trial "country" data.

## A.6 Ablations

**Table 7: Ablation experiments on removing different parts of the text from the input data. The first row denotes the complete input data. The rest of the rows indicates results from removal of that specific text feature only (preserves the other inputs text features).**

| Ablation | PR AUC | ROC AUC | F1 | Acc. |
|---|---|---|---|---|
| All Data Included | 0.766 ± 0.007 | 0.679 ± 0.005 | 0.726 ± 0.005 | 0.647 ± 0.005 |
| No Trial Summary | 0.703 ± 0.011 | 0.613 ± 0.007 | 0.000 ± 0.000 | 0.386 ± 0.007 |
| No Trial Tabular Data | 0.770 ± 0.008 | 0.684 ± 0.006 | 0.632 ± 0.005 | 0.616 ± 0.004 |
| No EC | 0.770 ± 0.008 | 0.683 ± 0.005 | 0.660 ± 0.005 | 0.625 ± 0.004 |
| No Drugs | 0.771 ± 0.008 | 0.685 ± 0.006 | 0.635 ± 0.005 | 0.618 ± 0.004 |

We conduct ablations by excluding text features from the trial summary, trial tabular data, eligibility criteria, and drug information, as shown in Table 7. Interestingly, removing certain text features marginally boosts ROC-AUC performance, but at the expense of F1 and Accuracy. Furthermore, excluding the trial summary significantly impairs LINT's performance, with all metrics dropping. This could elucidate why removing other features doesn't impact the results, as the model appears to predominantly rely on the trial summary.

---

[5]Top 10 sponsors: GlaxoSmithKline, Merck Sharp & Dohme LLC, Sanofi Pasteur, a Sanofi Company, Amgen, Pfizer, National Cancer Institute (NCI), Novartis Pharmaceuticals, Abbott, Bristol-Myers Squibb, Novartis Vaccines

Despite this, LINT might still find other outcomes important, yet it primarily hinges on the trial summary, thereby heavily influenced if it's absent. Future studies should delve into this dependency since other text features also significantly contribute to trial outcome prediction.

*A.6.1 Case Studies.* In this section, we take a closer look at 2 different biologic drugs. Secukinumab and Botulinum toxin type A, two random examples chosen from the test set, for the fair evaluation of LINT.

**Case 1: Secukinumab (NCT02404350)**[6] [34, 53]. First, let us take a look at a successful test prediction. As a brief summary, Novartis Pharmaceuticals sponsored this study, and the main goal of this study was to demonstrate efficacy on inhibition of progression of structural damage of secukinumab in subjects with active Psoriatic Arthritis (PsA) as measured by improvement in physical function measured by Health Assessment Questionnaire and skin and nail improvement for psoriasis signs. This study was deemed successful and had P-value of less than 0.0001 in its Estimation Parameter of Odds Ratio. LINT predicted success with a normalized score of 0.79 (from 0 to 1).

**Case 2: Botulinum toxin type A (NCT02660359)**[7] Here, we take a look at an incorrect test prediction. This study was sponsored by Ipsen, and the primary purpose of this trial was to analyze safety and efficacy of two Dysport [40] (Similar to Botox) doses (600 units U and 800 U), compared to placebo in reducing urinary incontinence (UI) in adult subjects treated for neurogenic detrusor overactivity (NDO) due to spinal cord injury (SCI) or multiple sclerosis (MS). This study was annotated to be unsuccessful due to the lack of participants; however, upon further analysis of the trial, all p-values were significantly less than 0.05, indicating statistical significance. This could indicate that if given more participants, a successful outcome may have been possible. LINT predicted success with a normalized score of 0.38 (from 0 to 1).

## A.7 Additional Figures and Tables

Figure 4 breaks down the histogram and density estimated distribution of completion years by phase and [small molecule drug vs biologics]. Table 8 shows a full example text input to LINT.

## A.8 Shap Values

Figure 7 shows the text Shapley (SHAP) value importance. The base value as shown in the upper left of the plot is the model logit output when the entire input text is masked. Using Shapley values [31, 32] from the package shap.readthedocs.io, we are additionally able to visualize the portions of the raw input text that affect model output the most (See Figure 7). The SHAP values additively explain the impact of unmasking each word on the model output–from the base value (where the entire input is masked) to the final prediction value (no mask). In short, the darker the color of the highlighted text, the more attention the model pays to it in its final classification. In short, Shapley values are the average marginal contribution of a feature value across all possible coalitions (combinations of features).

Features are masked out by random sampling from the existing dataset when not considered for traditional tabular feature classification. In this case, masking is done by replacing words with the [MASK] token by the SHAP package. Section A.6.1 shows an example of this. From this, we see that the model generally does rely on informative portions of text.

*A.8.1 Example Input.* Received 20 February 2007; revised 12 March 2009; accepted 5 June 2009

---

[6]https://clinicaltrials.gov/ct2/show/study/NCT02404350
[7]https://clinicaltrials.gov/ct2/show/results/NCT02660359

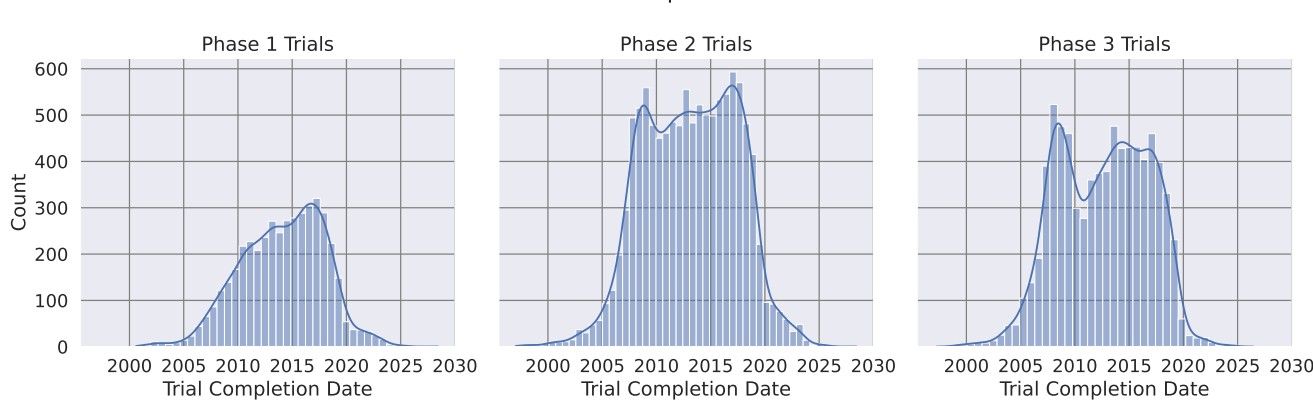

**Figure 3: All Trial Completion Years. Note that some completion years may be in the future due to projections. Completion Years by small molecule drugs and Biologics is broken down in Figure 4.**

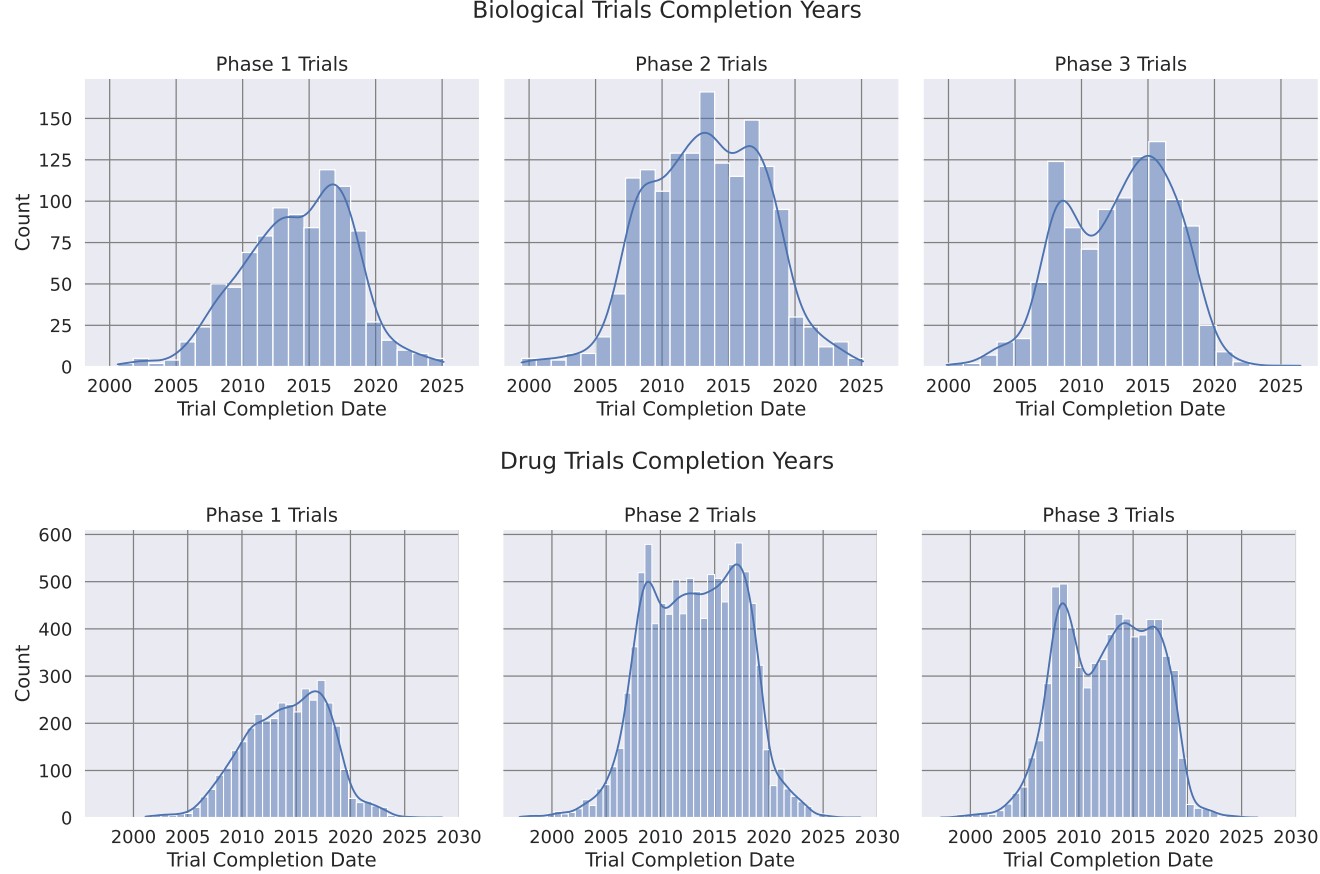

**Figure 4: Additional plots for the distribution of completion years by phase and modality. Histograms of trial completion years by phase for biologics and small molecule drugs**

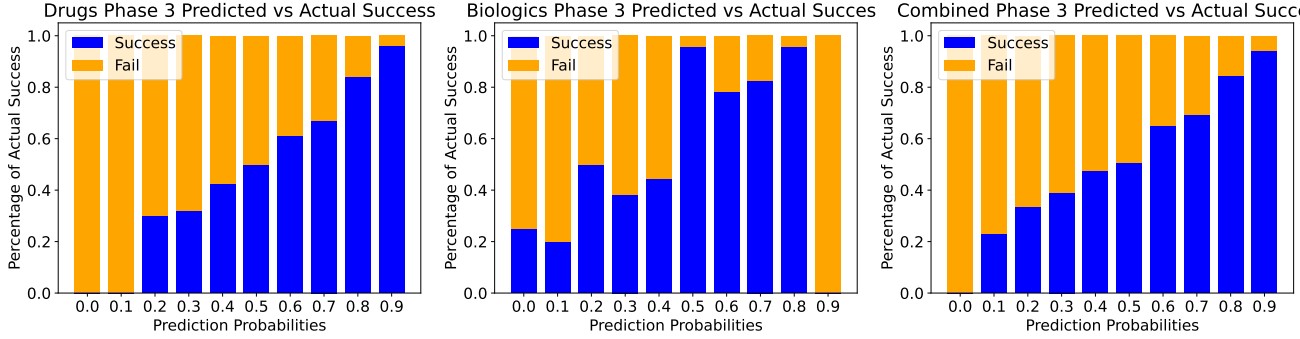

**Figure 5: Phase 3 Predictions of success vs actual success. This can be interpreted as LINT's predicted probability of success in the X-axis (e.g., X=0.1 contains all success probability predictions in the range [0.1,0.2)), versus the actual probability of successful trials among the predicted successful trial on the Y-axis (given the predictions ). Combined refers to the combined.**

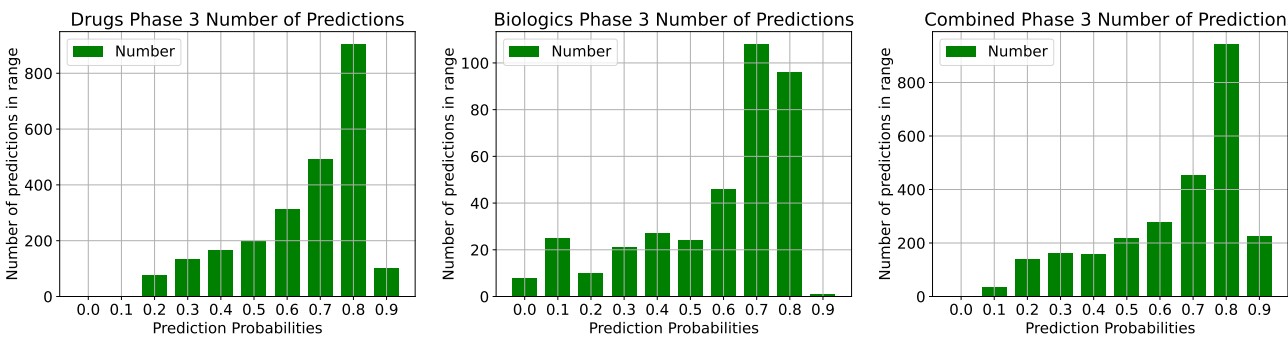

**Figure 6: Phase 3 Number of predictions. Combined refers to the combined. From these figures, we are able to see that many biologics classifications are made close to the .5 threshold, indicating that they are harder to classify.**

Secukinumab (Cosentyx) is a human monoclonal antibody designed for the treatment of uveitis, rheumatoid arthritis, ankylosing spondylitis, and psoriasis. Secukinumab is an interleukin-17A (IL-17A) inhibitor marketed by Novartis. IL-17 is a group of proinflammatory cytokines released by cells of the immune system and and exist in higher levels in many immune conditions associated with chronic inflammation. By targeting IL-17A, secukinumab has shown excellent efficacy in psoriasis by normalizing skin histology and was approved by the United States Food and Drug Administration on January 21, 2015 to treat adults with moderate-to-severe plaque psoriasis.

The purpose of this study is to provide confirmatory evidence of the safety and efficacy of two Dysport® doses (600 units [U] and 800 U), compared to placebo in reducing urinary incontinence (UI) in adult subjects treated for neurogenic detrusor overactivity (NDO) due to spinal cord injury (SCI) or multiple sclerosis (MS).

**Figure 7: An example plot of which parts of the text affect the output of the LINT model the most. This is a visualization of Shapley values https://shap.readthedocs.io of the text from 2 trials: https://clinicaltrials.gov/ct2/show/NCT02404350 on the top and https://clinicaltrials.gov/ct2/show/results/NCT02660359 on the bottom. The darker the color, the more the word affects the final output logits.**

**Table 8: An example of the text data (associated with https://clinicaltrials.gov/ct2/show/NCT00000172) that is input to the PLM. Each row represents one string. The titles are bolded. Note that for this particular trial, there is only one drug that it considers. If there are $n$ more drugs, this table would have $5 * n$ more paragraphs accordingly.**

---

**Trial Text**

Galantamine is an experimental drug being evaluated in the United States for the treatment of Alzheimer's disease. Results from previous clinical trials suggest that galantamine may improve cognitive performance in individuals with Alzheimer's disease [29]. It is not a cure for Alzheimer's disease. Nerve cells in the brain responsible for memory ...

**Trial Eligibility Criteria**

inclusion criteria probable alzheimers disease minimental state examination mmse 1022 and adas greater than or equal to 18 Alzheimers disease assessment scale cognitive portion adascog11 score of at least 18 opportunities for activities of daily living caregiver subjects who live with or have regular daily visits from a responsible caregiver ...

**Additional Trial Info**

eligibility gender all eligibility min age child eligibility max age none allocation randomized intervention model parallel assignment primary purpose treatment masking double sponsors small location countries north america ...

**Drug Description**

Galantamine is a tertiary alkaloid and reversible competitive inhibitor of the acetylcholinesterase AChE enzyme which is a widely studied therapeutic target used in the treatment of Alzheimers disease.A1018 First characterized in the early 1950s galantamine is a tertiary alkaloid that was extracted from botanical sources such as Galanthus nivalis.A201968 Galantamine ...

**Drug Pharmacodynamics**

Galantamine is a competitive and reversible inhibitor of acetylcholinesterase that works to increase acetylcholine levels.L13571 Galantamine acts both centrally and peripherally to inhibit both muscle and brain acetylcholinesterase thereby increasing cholinergic tone.A201968 Galantamine is also a positive allosteric modulator of neuronal nicotinic acetylcholine receptors.A1022A201968 As dementia is a progressive neurodegenerative ...

**Drug Toxicity**

The oral LDsub50sub of the active ingredient galantamine hydrobromide in rats is 75 mgkg.L13709 Symptoms of overdose are expected to be similar to those of cholinomimetics which involve the central nervous system the parasympathetic nervous system and the neuromuscular junction. Effects of a cholinergic crisis include severe nausea vomiting gastrointestinal ...

**Drug Metabolism**

In vitro study findings suggest that about 75 of the drug is metabolized by CYP2D6 and CYP3A4. CYP2D6 promotes Odemethylation of the drug to form Odesmethylgalantamine and the CYP3A4mediated pathway forms the galantamineNoxide.A182993 Important metabolic pathways also include Ndemethylation epimerization and sulfate conjugation.A203444 Other metabolites include norgalantamine Odesmethylgalantamine Odesmethylnorgalantamine epigalantamine ...

**Drug Absorption**

Over a dose range of 832 mgday galantamine exhibits a doselinear pharmacokinetic profile. The oral bioavailability of galantamine ranges from 90100. Following oral administration the Tmax is about 1 hour.L13571 Following 10 hours of administration the mean galantamine plasma concentrations were 8297 gL for the 24 mgday dose and 114126 ...