# OpenReview forum: "LINT: LLM Interaction Network for Clinical Trial Outcome Prediction"
_KDD.org/2024/Workshop/AIDSH — KDD-AIDSH 2024 Poster_

### Official Review · Reviewer_9s72 · 2024-06-11
**Review for Paper #3**

**Rating:** 7
**Confidence:** 4

**Review:**

**Summary:**
The paper introduces the Language Interaction Network (LINT), a framework designed to predict clinical trial outcomes using free-text descriptions, particularly for biologics lacking traditional molecular data. LINT leverages pretrained language models and a graph attention model to process trial descriptions, drug properties, and disease codes.

**Pros:**
- The writing and presentation are nice and clear, easy to follow.
- Transparency: the code is also clear.
- The significance of this work is substantial, as illustrated in the introduction.

**Cons:**
- It would be better to include a model architecture figure.
- The GRAM Encoder could be written in the preliminaries. The phrase "See Section 2.2" is also in Section 2.2.
- How is the validation set split? The table shows the partitions of the training set and test set, but a more detailed validation set split strategy should be included.
- Though the paper claims it beats the latest baselines, are LR, SVM, DT, AdaBoost, and RF the latest baseline methods?

---

### Official Review · Reviewer_Wp5T · 2024-06-15
**Review for Paper #3**

**Rating:** 5
**Confidence:** 4

**Review:**

Summary

The author proposed the Language Interaction Network (LINT) method. This method utilizes natural language processing techniques and only uses the text descriptions of clinical trials for prediction. Specifically, LINT consists of two modules: one is a text encoder based on the pre-trained language model for processing text data; the other is a disease encoder based on the graph attention mechanism for processing disease codes. The outputs of these two modules are concatenated and input into a multi-layer perceptron for binary classification.

Advantages:
1. A new method for predicting the clinical trial results of biologics has been proposed, addressing the shortcomings of existing methods in this regard.
2. The experimental results show that this method has good performance in predicting clinical trial results.
3. The method is simple and feasible, and easy to expand and apply.

Disadvantages:
1. There is a lack of a framework diagram in the manuscript to help readers understand the structure of LINT.
2. Insufficient novelty. The performance improvement of this method is merely achieved by using more modalities of data. The processing of each modality is based on existing methods, and the interaction between modalities is a simple concat, lacking more novel technical contributions.
3. The baseline models for experimental comparison are very strange. The author mentions in the paper, "Each baseline method is upgraded with the same input embedding used in LINT", which means the difference between the baseline models and LINT is only the final classifier (i.e., the MLP of LINT). The author should compare with more related works, such as Doctor2Vec and DeepEnroll mentioned in the manuscript.

Other issues:
1. The author should provide the full name of GRAM when it first appears in the manuscript.
2. Some double quotation marks have formatting problems, such as in lines 237 and 300.
3. The author needs to maintain the consistency of wording in the manuscript. For example, the "Mode" column in Tables 2, 3, 4, and 5 uses completely inconsistent writing styles.

---

### Decision · Program_Chairs · 2024-06-28

Accept (Poster)